# Axiology of Cultured Meat and Consumer Perception: An Analysis Based on the Phenomenology of Perception

**DOI:** 10.3390/foods15010034

**Published:** 2025-12-22

**Authors:** Béré Benjamin Kouarfaté

**Affiliations:** Departmental Unit of Management Sciences of Lévis, Lévis Campus, University of Quebec at Rimouski, 1595 Bd Alphonse-Desjardins, Lévis, QC G6V 0A6, Canada; berebenjamin_kouarfate@uqar.ca

**Keywords:** phenomenology of perception, axiology of consumption, axiology, perception of cultured meat, perception of artificial meat, perception of in vitro meat, food phenomenology, axiology and communication

## Abstract

This study presents a systematic literature review to examine how the axiological values associated with cultured meat influence consumer perception, using the phenomenology of perception as an analytical framework. Fifty-four peer-reviewed qualitative and quantitative studies, identified through the Libraries Worldwide database, were analyzed using NVivo 12 software, based on predefined keywords and a rigorous selection grid. The results highlight several groups of axiological values that shape consumer attitudes, including the previously unexplored “axiological value of co-production” of cultured meat. Specifically, “dogmatic co-production” (e.g., religious or cultural co-production) appears to significantly enhance consumer perception and acceptance of cultured meat. The main limitation of this study lies in the absence of primary phenomenological field data, which may introduce researcher subjectivity inherent in qualitative paradigms. Nevertheless, the use of existing empirical studies ensures the relevance and reliability of this review. This research offers practical implications for communication strategies, suggesting that aligning messages with key axiological values and their amplifiers, particularly those related to co-production, can strengthen trust in and acceptance of cultured meat. For industry stakeholders, these findings provide guidance for value-driven positioning aimed at increasing consumer confidence. Academically, the study offers a novel perspective by integrating axiological analysis with phenomenology in the context of food technology adoption. Socially, it helps identify consumer concerns and expectations regarding the axiological values perceived as essential for the acceptance of cultured meat. The study’s originality lies in its application of phenomenological analysis to axiological frameworks and in highlighting the central role of co-production, particularly dogmatic co-production, in shaping consumer perception.

## 1. Introduction

Cultured meat is known by several names. For example, in vitro meat (IVM), artificial meat, or cultured meat all refer to the same type of meat. Several studies have presented cultured meat as the most realistic and sustainable solution to the problems of current meat production and consumption [1,2], making it possible to meet the growing needs of humanity [3]. According to several authors, cultured meat could provide sustainable solutions not only to ecological problems, but also, and especially, to the current and future nutritional problems of the world’s population [1,3,4]. Other studies have shown that these problems will increase proportionally to the growth of the world’s population, which, according to the UN in 2024, will reach 10.3 billion inhabitants by 2080 [5]. Similarly, individual incomes will also experience a sharp increase, leading to a rise in meat consumption [3]. This high demand for meat among the population results in increased intensive livestock farming and thus an increase in greenhouse gas emissions. Furthermore, several studies show that the production of animal feed also contributes significantly to deforestation, the massive use of natural resources, and climate change [1].

However, several previous studies, such as those by [1,6,7], have concluded that cultured meat is more environmentally friendly, animal-friendly, healthier, and safer than conventional meat. This is why cultured meat is considered an environmentally sound solution for the planet [1,7].

Given these invaluable benefits, cultured meat is generating increasing interest in the research community. To fully understand the advantages and value of cultured meat, on the one hand, and the impact of these values on consumer perception, on the other, a phenomenological study of axiological values appears promising, in accordance with the work of [8]. Indeed, according to these authors, an epistemological analysis reveals that a phenomenological study can provide an empirically and methodologically rigorous understanding of consumption patterns [8]. A phenomenological analysis, or the phenomenology of perception, is a study of the structures of subjective experiences and consciousness [9,10]. It explores and describes the meaning individuals ascribe to their experiences, without prejudice or preconceived theories, in order to grasp fundamental structures and meanings [11]. For example, in this study, phenomenology would allow for the observation of the cultured meat phenomenon through literature, from production to consumption, including the distribution channels of artificial meat. This type of study is part of the ongoing struggle to claim its rightful place in contemporary research, and the relevance of this study lies in its interdisciplinary investigations of the brain and mind [9,11]. Moreover, several authors believe that “phenomenological inquiry should be considered a vehicle and method for accessing the food experience” [9], as is the case with this research, which aims to understand the consumption behavior of cultured meat. Furthermore, studies of consumer perception are still lacking, either from an axiological perspective (an analysis based on the fundamental values of cultured meat) or through the phenomenological method. Yet axiological studies [12,13] have also proven effective in consumption studies in general and food consumption studies in particular. Indeed, axiological value refers to the qualities that confer value on something or someone [14]. Axiology is a science originating from philosophy that studies the nature, origin, and different types of values (moral, aesthetic, epistemic, etc.) and examines the nature of what is good or worthy of esteem. It allows us to understand how these values are measured and compared [15]. Axiological studies can also be defined as a science of values (what has value, what can be the subject of a value judgment) [16]. In the context of cultured meat, for example, the advantages identified by several authors [2,17,18] namely, food security, animal welfare, environmental protection, and the nutritional contribution of cultured meat can therefore be linked to this notion of axiological value. Furthermore, the adoption of responsible consumption, the development of vegetarianism and veganism, and ethical behaviors that promote cultured meat are thus aligned with the axiological values of cultured meat. This is why it is important to understand the impact of axiology on consumers’ perceptions of cultured meat through a phenomenological analysis of perception.

## 2. Literature Review

Climate and environmental changes have led researchers to explore various solutions to address these problems. One such solution is the production and consumption of cultured meat as a replacement for conventional meat from traditional livestock farming [1,2]. Indeed, several studies have shown that cultured meat could meet humanity’s growing needs while reducing greenhouse gas emissions associated with conventional meat production [1,3]. Authors such as [3,19] define cultured meat as a product or outcome obtained by culturing cells in a laboratory using regenerative medicine techniques, starting with stem cells previously harvested from an animal [3,19]. Therefore, it is not a meat substitute, but rather a healthier, more sustainable, or more environmentally friendly meat that utilizes methods to mitigate the effects of climate change [19]. However, consumer perception of this meat remains largely negative. For example, in Canada, cultured meat (as an innovation) faces significant obstacles regarding its adoption and broader social acceptance [2,20,21,22]. Indeed, a survey conducted in Canada in 2020 revealed that only 22% of the population said they were willing to try it [20]. In this context, public perceptions of cultured meat are diverse [4]. Thus, understanding the axiological values of cultured meat could improve consumer perceptions.

Moreover, several studies in the literature have explored consumer perceptions and/or attitudes toward cultured meat. For example, [4,18,23,24] identified the determinants (ethical, intrinsic, informational, and belief-based) that influence consumer attitudes toward this meat. In their work, [25] reviewed the challenges encountered and the prospects surrounding cultured meat. Other researchers have explored various topics related to this meat, including: the comparison of different meat alternatives (plant-based meat imitation, insects, aquaculture, etc.) and consumer perceptions [26], eco-emotions and their influence on attitudes [22], factors that positively or negatively influence consumption intention [27,28,29], the likelihood of trying, buying, paying for, and consuming [30], culture-based purchase intention [28,31,32], and the influence of received information on purchase intentions [33,34]. Others have studied consumer perceptions of cultured meat according to their diet [35], and according to their place of residence and level of education [36].

However, these studies did not explore the impact of the axiological values of this meat on consumer perceptions. Indeed, axiology is a Greek word equivalent to “axia” or “axios,” meaning “value” or “quality.” Thus, axiology can be defined either as a science of sociological and moral values, or, in philosophy, as “both a theory of values (axios) and a branch of philosophy concerned with the domain of values.” The works of [37] on the genealogy of morality and those of [38] have shown that axiology “must be considered as a search to establish a hierarchy among the values” of an object of study, arguing that it could be composed of two parts: ethics and aesthetics. For several authors, these two branches of axiology highlight the values that are both distinct from each other and common to the same object of research [38]. Based on axiological philosophy, it is shown that the two branches of axiology (ethics and aesthetics) of a subject of study are two axiological domains, each referring to the “world of values.” In other words, ethics and aesthetics should be “subject to the necessity of being addressed in terms of value, beginning with the most general: ‘good,’ ‘evil,’ ‘beautiful,’ ‘ugly,’ and others” [39,40]. Furthermore, these two areas are not the only avenues of analysis, as the question of axiology in its rhetorical aspect was the subject of the philosopher Perelman’s work [41]. Very often, the notion of axiology is used within the logic of value systems with the aim of objectivity. However, the expression “axiological neutrality,” advocated by several authors, aims to support a viewpoint by which maximum objectivity can be achieved without using value judgments or critiques of the object of analysis [40]. This understanding would imply ignoring the existence of a hierarchy between the ethical and aesthetic values of an object of study. In any case, there are several other values that can be taken into account in an axiological study. In fact, several authors have established a list of values called the “register of axiological values” and have determined that each object of study refers to a certain type and number of values that correspond to a specific object of study [42,43]. For example, in the register of axiological values, authors have grouped a number of value categories into a dozen categories: civic, domestic, economic, epistemic, aesthetic, ethical, sensory, functional, hermeneutic, legal, pure, and reputational [14,15,40]. These different values vary according to the type of object and product. But several authors agree that aesthetic and ethical values remain permanent, regardless of the type of object [14,15,40].

In the literature, recent studies have explored various themes from an axiological perspective. For example, Golev [44] studied the cognitive and communicative axiology of modern virtual educational communication. The results of his study show that online pedagogy is perceived with caution or mistrust by many participants in pedagogical communication, and in some cases negatively. Similarly, Verbeeck-Boutin [40] suggests constructing a value grid, or more precisely, a conceptual model of sets of relationships between these values. Based on works of art, he believes that such a model would help to objectify choices and foster interdisciplinary discussion before decision-making [40]. Thus, for each axiological study, it is advisable to establish a list of the axiological values of the object of study. For example, in studying the axiology of group identity among bikers based on internet communication, Dankova et al. [15] highlighted the values that govern the group identity of bikers. The authors argue that the social category of “biker” is constructed through the values of “motorcycle,” “pleasure,” “danger,” and “brotherhood” [15]. Similarly, Rykała and Żołnierczuk [45] also studied the axiology of communication space and contemporary trends in the design of a transport hub. In this axiological study, they focused primarily on the aesthetic values of three transport hubs, highlighting the character of these places without emphasizing their ethical value [45]. Therefore, within the framework of this study, a register of values related to cultured meat will be established. Thus, the objective of this study will be twofold: first, to establish a register of the axiological values of cultured meat, and second, to understand the impact of these values on consumer perceptions and their acceptance of cultured meat.

To understand these consumers’ perceptions, an analysis of previous studies based on the phenomenology of perception will be conducted. This analysis will allow us to observe, analyze, and understand these perceptions throughout the entire cultured meat value chain (e.g., identifying the axiological values from production to consumption, including distribution). Indeed, according to [9], phenomenology is the study of the structures of subjective experiences and consciousness. It is considered a vehicle and method for accessing the food experience [9]. For example, using interpretive phenomenology, Dibsdall et al. [46] conducted their study to understand the experiences and beliefs related to food among a small group of low-income British women [46]. In this work, the authors suggested that this type of study lends itself well to in-depth qualitative methods and recommended that data be collected through interviews in a phenomenological study [46]. Therefore, the phenomenology of perception is seen as a methodological science [47,48] ideal for consumer studies. This is why several authors such as [11,47,48,49] have used this method in their various research on consumer consumption and perceptions. For example, in his exploratory phenomenological study of the lived experience of vegans or vegetarians in a predominantly non-vegetarian society and culture, Edwards ([11], p. 111) argues that “the use of phenomenological inquiry is well suited to uncovering the lived experience of this phenomenon in a way that no other method of inquiry could” [11]. In this study, the author also drew on his own personal experiences, which allowed him to uncover the phenomenon while supplementing the data. He highlights themes based on the use of Hyppolite’s [50] and Bachelard’s [49] “inside versus outside” descriptions. Using this methodology, Weiss [47] showed that various qualities become materialized in wines and that this process of materialization can be understood under very different regimes of social activity and meaning. Similarly, based on this methodology, neurobiological studies conducted among individuals suffering from loss-of-control binge eating (LCBE) consistently show signs of addictive alterations, including hyperactivity of reward centers, stress reactivity, and cognitive impairment, as well as maladaptive use of ultra-processed foods [48]. Ningtyias and Kurrohman [51] conducted a study on the phenomenology of food taboos and recommended foods for pregnant women in Pendhalungan society. They used a phenomenological approach to describe unique consumption patterns. This is a content analysis method [51]. According to Merleau-Ponty, the phenomenology of perception involves perceiving objects in a holistic manner, taking into account the perceived world (the environment), the perceiving body, and the consequences of this new approach for the eternal questions of the cogito, temporality, and freedom. It therefore suggests observing objects or phenomena in a broader way.

Thus, in the context of this study, consumer perception will be analyzed globally. This holistic view of perception is appropriate for this study because it not only makes it possible to understand consumers’ perception of cultivated meat, but also their perception of the environment (the perceived world). Specifically, the phenomenology of perception allows us to observe and analyze consumer perceptions from production to consumption. The advantage of this type of research is that it allows us to identify the factors influencing consumer perceptions and attitudes at each stage of a product’s journey, from its creation to its final use. Thus, through this study, the axiological values of cultured meat likely to influence consumer perceptions will be observed and analyzed through the literature, following its path from production to consumption via the distribution chain.


**Conceptual Framework:**


The theoretical framework below highlights the relationship between the axiological values of cultured meat (ethical, aesthetic, utilitarian, reputational, hermeneutic, legal, commercial, or economic), their amplifiers, and the perception and acceptance of this meat by consumers. This conceptual framework was defined following a literature review. This relationship is illustrated in Figure 1. Based on this framework, the following research hypotheses were formulated.

**H1.** 
*The ethical axiological value of cultured meat positively influences consumer perceptions.*


**H2.** 
*The aesthetic axiological value of cultured meat positively influences consumer perceptions.*


**H3.** 
*The utilitarian axiological value of cultured meat positively influences consumer perceptions.*


**H4.** 
*The reputational axiological value of cultured meat positively influences consumer perceptions.*


**H5.** 
*The hermeneutic axiological value of cultured meat positively influences consumer perceptions.*


**H6.** 
*The legal axiological value of cultured meat positively influences consumer perceptions.*


**H7.** 
*The commercial axiological value of cultured meat positively influences consumer perceptions.*


**H8.** 
*The amplifiers of the axiological values of cultured meat positively influence consumer perceptions.*


**H9.** 
*There is a positive relationship between consumer perception and their acceptance of cultured meat.*


**Figure 1 foods-15-00034-f001:**
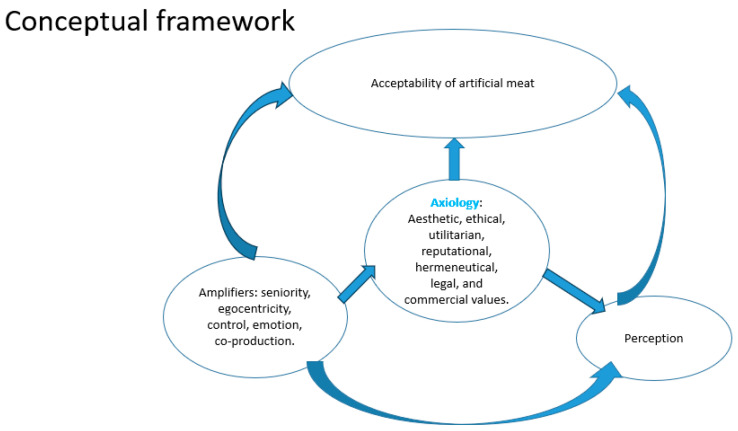
Conceptual Framework.

## 3. Material and Methods

The objective of this systematic literature review is to identify, appraise, and synthesize empirical studies that address the axiological values associated with cultured meat. The paper further seeks to outline a coherent research agenda for the management science literature. The methodological approach follows the protocol of Tranfield et al. [52], comprising the formulation of the research question, the identification of relevant studies, the assessment of study quality, the selection of eligible articles, the synthesis of evidence, and the interpretation of findings. Similarly, the methodology of this article followed the “PRISMA 2020 guidelines” in accordance with the elements of the PRISMA checklist.

### 3.1. Formulation of the Research Question

The research question addressed in this article is to propose a holistic conceptual framework for understanding the effect of cultured meat axiology on consumer perception and acceptance, using a phenomenological analysis approach. This involves a systematic, multidisciplinary literature review. To this end, several article selection criteria, such as inclusion and exclusion criteria, were applied, as shown in Table 1. The use of these criteria ensures compliance with the conditions established by [52].

### 3.2. Identification of Relevant Publications

A thorough search was conducted across a wide variety of bibliographic databases using several keywords. Table 1 presents the selection criteria, while Table 2 details not only the different keywords but also the various stages of selection and filtering of these studies. It should be noted that this is a literature search covering all years up to August 2025. The keywords used are: **Phenomenology of Perception, Axiology of Consumption, Axiology, Cultured Meat and Perception, Artificial Meat and Perception, In Vitro Meat and Perception, Phenomenology and Food, and Axiology and Communication**. These keywords were identified and used based on the study by [21], which examined the nomenclature of cultured meat and its impact on consumer acceptance, as well as on the concepts of perception, axiology, and phenomenology.

### 3.3. Evaluation of the Quality of the Selected Studies

Following a step-by-step evaluation, 54 studies were deemed relevant. Based on the article quality assessment criteria recommended by [52,53], only peer-reviewed articles were included in this review. Since the objective of this research is to understand and highlight the impact of axiological values on consumer perception through a phenomenological analysis, and thus on the acceptance of cultured meat, studies from all research disciplines were included.

The documentary research was carried out through the Sofia library and identified 354 articles. This is the “unified” library of all universities across all Canadian provinces. This library contains all databases, including Web of Science and Scopus. For the purposes of this study, all these databases were taken into account (including Web of Science and Scopus) in order to include scientific studies from all fields and disciplines. Likewise, the articles were selected automatically using the “advanced search” function of this same Sofia library by automatically selecting the “Libraries worldwide” option.

In terms of queries and article filtering, automatic filtering was applied using Sofia’s search options. For example, for databases, by selecting the “all databases” option; for article quality, by selecting the “peer-reviewed articles” option; for duplicates, by selecting the “remove duplicates” option. For “content type,” the “Full text” and “Scholarly publication” options were selected. For publication year, the “all years” option was selected, up to 30 August 2025.

The retrieved articles were jointly evaluated by the entire team consisting of two specialists in qualitative data analysis and the author of this article. Disagreements were resolved by consensus; for example, by evaluating the methodology, study design, or results. The decision to include all articles from all disciplines reflects the multidisciplinary nature of this article. To ensure sufficient quality of the studies reviewed, only peer-reviewed articles were retained. Indeed, they undergo rigorous academic review processes that ensure methodological rigor, follow standardized methodologies that facilitate comparison and synthesis, and are generally less exposed to commercial or political bias than other types of literature such as industry reports or policy documents. These quality criteria have already been used by other researchers, including Kouarfaté and Durif [2], Abd Hanan et al. [24], and Ji and Lee [54]. Moreover, to ensure accessibility, quality, and relevance, only publications written in English and French were considered. A total of 54 studies were retained after this process.

### 3.4. Selection of Studies

The keywords used in this work were applied precisely to the search in all fields, and then to all article titles appearing in English and French. The research protocol used in the context of this article is represented by the “PRISMA 2020 flow diagram for new systematic reviews which included searches of databases and registers only” (see Figure 2). This analysis demonstrates the quality of the studies included in the systematic review and allows for the organization of this review according to different themes (axiological values). Several elements were introduced to select and filter these articles, as shown in Table 1. The filters used in Table 1 are: peer-reviewed journals, exact wording of keywords in the title, and searches of journals through abstracts and full texts. Finally, duplicates were eliminated. Table 2 presents the different stages of selection and filtering of these studies.

### 3.5. Data Analysis and Interpretation

The analysis of the 54 articles was conducted in two stages:

(1) The first involved a quantitative analysis based on the categorization system described by Armat et al. [56] and Wendt and Weinrich [57]. This categorization system takes into account information such as the authors’ names, the year of publication, the main research questions, the research methods used, sample size, the disciplines or sub-disciplines of publication, the arguments put forward, and the research strategy. These elements were recorded in Microsoft Excel. The information extracted from the articles was entered into Microsoft Excel and then summarized in Appendix A, Table A1. In addition, the collected data were descriptively analyzed, and the results are presented in Section 4.1.

(2) The second analysis was carried out qualitatively. The 54 articles were coded in order to identify those highlighting at least one concept related to phenomenology, consumer perception, the axiology of consumer goods, and the values of cultivated meat. This involved analyzing existing qualitative and quantitative studies in the literature and in library databases worldwide, using NVivo 12 software. Table A1 (in the Appendix A) presents only the main studies related to these research themes. The selected articles were jointly analyzed by the entire team, which consisted of two specialists in qualitative data analysis and the lead author of this article.

Disagreements were resolved by consensus through an assessment of the methodology, study design, or results. A priori thematic coding was carried out. In line with the recommendations of Flemming et al. [58], thematic analysis is the best approach for a systematic literature review integrating mixed studies, as it allows for the identification of codes and overarching themes that recur throughout the data [59]. These various themes provide a means of understanding the overall meaning conveyed by the articles [60]. Since the themes were developed a priori, this implies that they were coded using the pre-existing coding framework found in the literature [61,62], as opposed to a posteriori coding based on an inductive categorization approach. Concretely, the a priori themes identified stem from the register of axiological values presented in the literature. The initial coding themes therefore consist of the twelve axiological values identified in the literature, namely the civic, domestic, economic, epistemic, aesthetic, aesthesic or sensory, ethical, functional, hermeneutic, legal, pure, and reputational registers [14,15,40].

Based on the analysis of the selected articles, the themes were developed through a rigorous process consisting of coding the meaning of a specific section of the text, and then iteratively organizing the open codes into themes (each theme designating either an axiological value or an amplifier of that value). Clearly, all twelve axiological values do not necessarily apply to each product, as the value register varies according to the type of object and product [14,15,40]. At the end of the coding process, only seven axiological values were actually coded.

Furthermore, to ensure inter-rater reliability and resolve potential discrepancies, the researchers rigorously adhered to their coding framework (a priori thematic coding). They then created and regularly (weekly) updated the chronological log of incidents or coding divergences between the researcher and the analysts. Since the themes and sub-themes were known in advance, the researchers were able to code according to the same themes and sub-themes, thereby reducing inconsistencies in the coding process [63].

## 4. Results

### 4.1. Results of the Descriptive Analysis (Profile of the Analyzed Articles)

Although the analysis of all articles took into account all publication years, it appears that these articles are relatively recent. Indeed, 65% of these articles were published after 2010 and two-fifths of them are less than five years old (see Figure 3).

They are primarily published in fields such as: Marketing, social science and consumption (18/54), Philosophy/anthropology (13/54), Nutritional science and meat science (5/54), Psychological and psychiatric science (3/54), Medical and food science/Consumption (3/54) (see Figure 4). Almost all of the selected articles (51/54) are published in journals included in the ranking of scientific journals and countries, notably: 32 in the first quartile (Q1), 5 in the second quartile (Q2), 7 in the third quartile (Q3), and 7 in the fourth quartile (Q4), which testifies to the quality of these articles. The journals in which these articles are published are diverse, including: Journal of Consumer Research (2/54), Frontiers in Sustainable Food Systems (2/54), Appetite (2/54), Foods (1/54), Frontiers in Psychology (1/54), International Journal of Research in Marketing (1/54), and Journal of Consumer Research (1/54). Although almost all of these articles are published in an English-speaking context (45/54), some are published in French (9/54). In terms of analytical methods, the results show that 36/54 of these studies are qualitative (literature reviews and/or document analyses, interviews, and focus groups), 3/54 are mixed methods, and 15/54 are quantitative methods.

### 4.2. Register of Axiological Values of Cultured Meat

The data analysis of this study allowed us to present the results through the lens of the axiological values of cultured meat, using a phenomenological methodology (observation of the cultured meat phenomenon, from production to consumption, including the distributors of artificial meat). Since no systematic literature review exists that explores all articles across the entire cultured meat value chain, based on the axiological values identified in the literature and adapting these values to cultured meat, the analysis identified seven axiological values of cultured meat and their amplifiers.

#### 4.2.1. Aesthetic or Intrinsic Value

For consumers, the aesthetic (or intrinsic) value of meat holds a significant place in their minds [64]. This aesthetic or intrinsic value thus influences consumers’ perceptions of meat. Aesthetics (phenomenology of perception and art theory) therefore plays a central role [64]. The intrinsic and aesthetic notion of cultured meat refers to its internal characteristics or attributes [31,65]. This is the case, for example, with sensory aspects: the shape, color, and tenderness of cultured meat [31,65]. Similarly, other sensory aspects such as texture [66,67] and the aesthetic appearance of cultured meat (its resemblance to conventional meat would positively affect consumer perceptions as well as their acceptance of consuming cultured meat [2,68,69]. For example, this analysis corresponds to the following excerpt from the results of the article by Kouarfaté and Durif ([2], p. 2748): “Perceived sensory quality is one of the variables listed by the authors. It comprises CM’s appearance, texture, flavor, taste, tenderness, sweetness and chemosensory attributes. Many other authors have also pointed to the impact of nutrients in artificial meat as an intrinsic factor influencing consumer attitudes”.

#### 4.2.2. Ethical or Extrinsic Values

The first axiological domain within the world of values is the domain of ethics. Indeed, every object of study presents its own values, highlighting ethical value or foundations based on moral values. Cultured meat is no exception to this principle, as several ethical questions arise concerning its production, distribution, and consumption. Several authors agree that the extrinsic or ethical dimensions of cultured meat are more important in its acceptance by consumers [70,71]. The ethical or sustainability concept then highlights several factors, such as the capacity to protect the environment and ensure the well-being of humans and animals following the production and consumption of this meat [2,33,68]. Regarding [72], it is the ethical value that encompasses the aspects of a social, ecologically sustainable, healthy and fair diet; and which would have an impact on the perception of cultured meat [72]. For example, here is an excerpt from the results of the article by Mancini and Antonioli ([68], p. 105) that underpins the coding of this theme: “Highly educated participants (current university students together with participants with Bachelor, Master and Ph.D. degrees) expressed a significantly higher appreciation of two intrinsic (safety and flavor) and two extrinsic (animal welfare and sustainability) attributes of cultured meat compared to less educated participants (those with primary to high school degrees). Seemingly, participants familiar with cultured meat showed a more positive attitude towards cultured meat, particularly concerning its intrinsic attributes and one extrinsic attribute (i.e., sustainability) when compared to those lacking any familiarity with the topic”.

#### 4.2.3. The Utilitarian Value of Cultured Meat

Utilitarian value refers to the usefulness a good or service provides to a user through its use or functional aspects. In the context of cultured meat, its utilitarian value can be associated with certain sensory aspects such as flavor, taste, and nutritional contribution [2,68]. Indeed, previous studies have listed characteristics that refer to intrinsic factors of cultured meat, such as [65] for nutritional quality, [31] for taste, [66,67] for appeal and taste. The coding of this theme (utilitarian value of cultured meat) is supported by the following quote from the excerpt of the article by Laestadius and Caldwell ([66], p. 2465): “Rather than suggesting that IVM is necessary in light of challenges such as climate change and population growth, it may be preferable to cast IVM as a product that stands on its own merits regardless of external circumstances. Traits relevant to public health nutrition such as nutritional content, taste and food safety then become particularly important.”

#### 4.2.4. Reputational Values/Stakeholders

Reputational values are the fundamental principles perceived by stakeholders that contribute to the overall value of an organization, product, or person. They encompass aspects of governance, social responsibility, and excellence, which strengthen trust and collective esteem. Managing them is crucial because a strong reputation can become a competitive advantage, while a damaged reputation can generate significant risks. For example, analyzing the entire value chain of cultured meat highlights the importance of stakeholders (producers, distributors, retailers, consumers, etc.). Indeed, several studies suggest that consumer perceptions and acceptance of cultured meat could be influenced by stakeholders [2]. For example, [73,74] suggested that stakeholders play an important role in the acceptance and marketing of new products or services, including cultured meat, which may be associated with GMOs (Genetically Modified Organisms). Furthermore, studies by Böhm et al. [6] assert that if stakeholders accept cultured meat, its reputation and consumer acceptance could be improved. Here is a coded excerpt from the article by Böhm et al. ([6], p. 222) suggesting the normative influence of cultured meat stakeholders on consumer perception: “The normative force of the innovators’ vision of in vitro meat was also splitting the interviewees. For some of them, in vitro meat is the better meat (Section 3.1) and even a step towards animal liberation (Section 3.2). Others see this so-called technological win-win solution running the risk of becoming first an elitist and then a mass product suspected to cheat the consumers”.

#### 4.2.5. Hermeneutic Values

This value is linked to consumers’ interpretation of cultured meat. In previous studies, for example, consumers often associated this meat with the notion of novelty. For Siegrist and Sütterlin [75] and Weinrich et al. [28], in addition to novelty, the perceived naturalness of cultured meat is also a determining factor. This interpretation can also be influenced by the initial information consumers receive about cultured meat. Indeed, several studies have suggested that the initial information consumers receive elicits their initial reactions to cultured meat [18]. The way a new product is marketed influences these initial reactions. These are particularly important for food products [27]. Similarly, Tuorila and Hartmann [76] emphasized the importance of consumers’ first impressions, arguing that they generate food curiosity (which corresponds to the consumer’s desire to learn more about the product’s production or processing methods). Furthermore, curiosity enhances consumer perceptions since it is often linked to the consumer’s learning capacity [27] and also reflects their willingness to try or test a product [76]. For example, the code for this theme is supported by the following excerpt from the article by Hwang et al. ([27], p. 5): “The first reactions are important factors when consumers are confronted with novel foods. There seem to be two attitudes toward new food: food curiosity and food neophobia. Likewise, there are some findings that initial reactions, including curiosity and neophobia, could affect WTB alternative meat. Since the alternative meat market is new, consumers’ first impressions are important. Therefore, we use food curiosity and neophobia as criteria of the initial reaction”.

#### 4.2.6. Legal Values

The legal axiological value of cultured meat is based on the added value of regulating the cultured meat sector, from production to consumption, including the distribution chain. Indeed, previous studies suggest that consumer perceptions of cultured meat could be improved through strict regulation of this sector. For example, studies by and [2,18,77], suggested that regulation would increase consumer confidence and thus promote its acceptance. Furthermore, other studies [78,79] have estimated that “the genetic testing sector and the cultured meat sector require effective regulation to build consumer confidence and improve their perception” [78,79]. For example, the coding of the legal value of cultured meat is corroborated by the following excerpt from the work of Kouarfaté and Durif ([18], p. 10): “Indeed, tweets emphasizing curiosity, regulation, neophobia, and conspiracy in relation to cultured meat generate cognitive, affective, and conative reactions”.

#### 4.2.7. The Commercial or Economic Value

In their proposed register, several authors have mentioned the economic value of the product [14,15,40]. Marketing attributes, including brand, name, packaging, price, and the origin of the meat as indicated on labels, are important elements that can influence consumer perceptions of cultured meat [80]. Here is an excerpt from the work of Hocquette ([80], p. 168) that confirms the influence of commercial factors on consumer perceptions: “As a consequence of all these profound developments, the main factors which currently affect meat purchases and consumption are, in addition to sensory factors (mainly color, tenderness, and flavor), psychological factors (including cultural factors and lifestyle), guarantees of hygiene and safety, as well as marketing factors such as price, brand, and labels based on origin, safety, local production and ethical production (reviewed by Hocquette et al. (2013a); Font-i-Furnols and Guerrero, 2014)”.

#### 4.2.8. The Axiological Value Amplifiers of Cultured Meat

The value amplifiers of an object or product are numerous. In the heritage field, for example, age is a highly significant value amplifier, as is rarity. Age is defined by the criterion of how old the object is, whereas rarity is defined by the object’s unique character. The criterion of age is so important that the construction or renovation date is the first piece of information provided by specialists in works of art [14]. These various value amplifiers serve to increase the value of a product or object. For instance, a work of art or an old and unique (rare) collector’s vehicle will see their value rise [14]. Similarly, the value of an ordinary car will be influenced by its age. In this case, the more recent the vehicle’s year of manufacture, the higher its economic value. In all cases, for each product, there exist axiological value amplifiers specific to that product. As part of this work, a list of value amplifiers for artificial meat is proposed.

The analysis of the work of several authors such as [2,14,21,31,46,69,72,81,82,83] has made it possible to identify and list a set of axiological amplifiers of artificial meat. Indeed, the work of Heinich [14] and Kombolo Ngah et al. [84] made it possible to identify the “Egocentric Systems” amplifiers. The studies by Kouarfaté and Durif [2], Dibsdall et al. [46], and Ningtyias and Kurrohman [51] were used to identify the “Control Issues” amplifier. Regarding “Emotions,” it was the work of Heinich [14] and Kouarfaté et al. [22] that allowed their identification. Similarly, “The notion of novelty or seniority” was identified by the studies of Bryant and Barnett [21], Mancini and Antonioli [69], and Browning et al. [83], while “The notion of co-production” was identified by Fleischer [39], Verbeeck-Boutin [40], Marion [85], and Faure [86].
-Egocentric Systems

“Egocentric systems” are linked to the uniqueness of individuals and to the social worlds they inhabit. The egocentric theme encompasses several social and psychological concepts. According to the author, each of these concepts constitutes a distinct field of research. This includes, for example, the importance of life-course factors in consumers’ food choices. The factors mentioned are: health status, education, roles, geographical location, cultural traditions, the food system, resources, and others. These findings are consistent with those of [84], who suggest that consumers’ place of residence can enhance their perception of cultured meat. Thus, for this type of meat, egocentric systems may also be considered value amplifiers. This observation can be confirmed by the following excerpt from the results of the work of Kombolo Ngah et al. ([84], p. 10): “Analyses of the data pool for 5 countries in common between survey 2 and 3 showed that WTP differed significantly according to country and education (Figure 7; Table 3). Interactions were significant between country and age, country and income, and education and income (Table 3)”.
-Control Issues

“Control issues” describe the way in which perceptions of control may influence attitudes towards food and health [46]. Here, the notion of “control issues” refers to the principle that “individuals are less likely to worry about health hazards they perceive as having some personal control over, compared to hazards they consider beyond their control” [46]. Healthy food choices are therefore generally considered to fall under individual control and consequently generate little concern. The desire to maintain control over one’s diet would constitute a factor in an individual’s consumption decisions. Moreover, several authors confirm that the main fears related to food concern invisible toxic contamination, which can trigger changes in behavior or experience in relation to food [51]. The coding of this theme is supported by the following excerpt: “Finally, “control issues” described how perceptions of control influenced attitudes toward food and health.” ([46], p. 298).
-Emotions

For heritage objects, Heinich [14] demonstrated that consumers’ affect towards an object constitutes part of its axiology. According to him, this emotional value, or “affect,” is a reflection of shared values. These value amplifiers for a heritage object can be expressed through authenticity, presence, and beauty [14]. Moreover, eco-emotions are specific emotions associated with cultured meat. In their work, Kouarfaté et al. [22] showed that eco-emotions influence both perceptions of and acceptance towards cultured meat. Thus, the analysis of the following excerpt confirms the code for the “emotions” theme: “Insofar as the addition of eco-emotions allowed the extended TPB model to explain 90.3% of the variance in consumer intent, it is important to consider eco-emotions, particularly eco-depression, in future advertising messages. This means avoiding depressing messages emphasizing the harmful effects of climate change for example and preferring optimistic messages such as those advocating happy sobriety”. ([22], p. 259).
-The notion of novelty or seniority

The notion of seniority or novelty, previously mentioned in the case of works of art, could also apply to artificial meat. For example, the value amplifiers for this type of meat could include the level of prestige associated with the product, given its new and innovative character. Furthermore, food neophobia defined as the reluctance or distrust to consume or avoid new foods has been identified by several authors as a factor influencing consumers’ acceptance of cultured meat [21,69,83]. According to these authors, the perceived lack of naturalness, the spread of conspiracy ideas, a disgust reaction, and fear of unknown risks associated with the new food technology can also influence, and even reduce, consumers’ willingness to eat cultured meat. This result is consistent with the following excerpt: “Indeed, sensory tests are crucial for acceptance of a novel food, in particular for meat substitutes such as cultured meat, since consumers are not willing to compromise a great deal on the taste of meat substitutes and need the so-called “familiar flavor” to reduce food neophobia” ([68], p. 12).
-The co-production of value

The object or product (good, service, personality, or other) exists and possesses characteristics as well as a minimum value related to its use. The axiology of consumption is a theory of the forms of value [85]. However, considering that the consumer is a co-producer of (use) value implies that it is the consumer who brings value to the object. In other words, it assumes that an object or product (isolated from the consumer-co-producer) has no value. This theory suggests not only that the value of an object is real only when it is integrated into a practice or usage, but also that the consumer remains a regular or habitual producer. Thus, according to [85], “the emergence of the value of an object results from its interaction with a subject.”

Nevertheless, several authors such as [39,40,85] agree that co-production allows for taking into account the consumer’s concerns during the design or production of the object. However, co-production does not constitute value in itself, but rather what Verbeeck-Boutin [40] called an axiological amplifier or value amplifier. Indeed, these amplifiers are the set of factors that contribute to increasing the value of the object of study. For instance, involving consumers in the production of cultivated meat could improve its values (aesthetic, ethical, utilitarian, reputational, hermeneutical, legal, and commercial).

Practically, this study suggests that during the production of cultivated meat, the practices of each religious and/or cultural group should be incorporated in accordance with the laws and dogmas of these different religious or cultural groups. Indeed, meat consumption is part of religious practices: Orthodox Christianity, Judaism, Islam, Christianity more broadly, and others [86]. According to [87], for example, the consumption of cultivated meat is strongly influenced by religious considerations, notably the Jewish “kashrut” and the Islamic “halal” precepts. Similarly, the consumption of certain types of meat complies with cultural restrictions and dogmas (e.g., Hinduism: ahimsa, Sikhism, or Jainism) of certain groups [33,88]. Therefore, within the framework of this study, the analysis suggests a “dogmatic co-production,” meaning religious or cultural, to satisfy the expectations of all socio-cultural and religious groups regarding meat consumption. An excerpt from the article by Hamdan et al. ([87], p. 2202), which was coded and reinforces this notion of dogmatic co-production, is: “Cultured meat is one of the most promising new products created by human, in line with the development of science and technology in human history. As a Muslim, every product and invention must have an adjudication from the perspective of Islam, if is compliant with the requirements of Islamic law. Since the culturing of meat is a new invention that has never been discussed by classical jurists (fuqaha’), an ijtihad by contemporary jurists must look for and provide answers for every technology introduced, whether it meets the requirements of Islamic law or not”. Similarly, the suggestion of dogmatic co-production is also supported by this excerpt taken from the work of Faure ([86], p. 1): Thus, in addition to the well-known prohibitions concerning the consumption of certain animals (pork, shellfish, etc.) and the ban on consuming blood, another prohibition less studied by anthropologists nevertheless contributes to constructing the boundaries of Jewish dietary practices in an extremely rigorous and restrictive way: the prohibition against mixing meat and dairy foods in the course of the same meal.

Finally, the excerpt from the results of the work of Kouarfaté and Durif ([2], p. 2749) that follows: “The consumption of certain types of meat complies with the cultural restrictions and dogmas of certain groups. In this study, religious and cultural morality is identified as a positive factor, while religious or cultural conservatism is a negative factor for accepting artificial meat”, corroborates this notion of dogmatic co-production.

## 5. Discussion

The results of this study, based on axiological aesthetic values, suggest that the aesthetic (or intrinsic) value of meat occupies an important place in consumers’ minds [64] and influences both perceptions and attitudes. These findings are consistent with previous research [31,65]. The intrinsic value (or aesthetic value of cultured meat) appears to rely on sensory aspects such as shape, color, texture, and perceived tenderness of cultured meat [31,65], and positively affects consumer perceptions as well as their willingness to consume cultured meat [2,68,69]. However, the results of the study by Abd Hanan et al. [24] show that it is rather the attitudes and traits of consumers that have the greatest impact on their perception and intention to consume cultured meat.

Regarding the ethical or extrinsic axiological value of cultured meat, analysis of the results indicates that its ethical value impacts consumer perceptions and acceptance. These findings align with those of previous studies [70,71]. Likewise, studies by [18,33,68] suggest similar results, particularly regarding perceptions of environmentally sustainable and healthy diets. Moreover, ref. [89] showed that beef purchasers are more concerned with extrinsic quality indicators than intrinsic ones, believing that extrinsic factors positively influence the eating experience of beef, while credibility attributes are generally disregarded except for the meat’s expiration date [89]. However, these results do not corroborate those of Zerfu et al. [90], who found that cultural beliefs may affect nutrition under certain circumstances, such as during pregnancy. According to [2,66], the values of cultured meat are associated with environmental and public health motivations, unnaturalness, regulatory considerations, and neophobia, which influence consumer perceptions of this type of meat [2,66].

Cultural values, race, gender, ethnicity, financial situation, and aspects such as quality, variety, balance, and moderation are values that can influence consumers’ perceptions and behaviors regarding meat [91]. Additionally, according to Teklebrhan [92,93], religious and sociocultural taboos are key variables affecting meat preferences and consumption within a given population [92,93]. On the other hand, the work of Ji and Lee [54] suggests that labels communicating the naturalness of cultured meat consequently help improve consumer acceptance and perceptions.

Similarly, regarding utilitarian values, the study suggests a relationship between the utilitarian values of cultured meat and consumer perceptions. These results are consistent with previous studies, including Mancini and Antonioli [68,69] and Kouarfaté and Durif [2] for flavor, taste, and nutritional contribution, and Bryant et al. [31] and Bhat et al. [65] for nutritional quality and taste. However, information related to the intrinsic attributes and positive externalities of cultured meat must be combined with different approaches to further improve consumer perception and acceptance [69].

Although there is very little literature on the reputational values of cultured meat, particularly those assigned by stakeholders, Kouarfaté and Durif [18] and Böhm et al. [6] highlight the importance of stakeholders in shaping the reputation of cultured meat among consumers. These results contradict those of Abd Hanan et al. [24], who identified only other factors that may influence consumer perceptions, namely: their attitudes and traits, situational impacts, information and nomenclature, the properties of cultured meat, perceived risks and benefits, familiarity and awareness, as well as competition with other alternative proteins.

Regarding legal values, several studies have suggested that regulation of the cultured meat sector could increase consumer trust in cultured meat [77,78,79]. The results of this study therefore support the body of previous findings.

Hermeneutic values of cultured meat (related to perceptions of its naturalness and novelty) have also been studied, with results partially aligning with those of this study. Indeed, due to its similarity to conventional meat, cultured meat is perceived as natural [2,27,76], corresponding to its hermeneutic axiological value. However, food neophobia [28,31,69,76] represents a negative factor in consumer perceptions of cultured meat.

As with any product, the commercial or economic value of cultured meat influences consumer perceptions. For example, as confirmed by this study, the literature shows that brand, name, packaging, price, and origin of meat can improve consumer perceptions [14,15,40,80].

Finally, in terms of amplifiers, egocentric systems associated with the consumer’s state, the need for control over one’s diet [46,51], consumer emotions [14], and even eco-emotions [22] act as amplifiers of identified axiological values, influencing consumer perceptions. Although studies have not investigated the effect of co-production of cultured meat on consumer perceptions, the results of this study suggest that co-production, particularly “dogmatic co-production” (religious or cultural), could meet the expectations of consumers from each sociocultural and religious group regarding laws and dogmas.

## 6. Research Avenues

Within the framework of this paper, one potential research direction would be to deepen this analysis by conducting a phenomenological study of consumer behavior regarding cultured meat in the field through qualitative analyses. Another avenue for research would be to measure the impact of each axiological value amplifier on the acceptance of cultured meat. A further study could focus on understanding the influence of “dogmatic co-production” by culture and religion on consumer perceptions of cultured meat.

It would also be of interest to carry out a study to determine the hierarchy among the various axiological values identified in this review. Research on the most effective communication methods based on axiological semiotics could be explored, as phenomenological studies suggest that messages are appropriated through the consumer’s different senses [94]. An epistemological analysis reveals that existential phenomenology can provide an empirically grounded and methodologically rigorous understanding of consumption phenomena [8]. In phenomenological analysis, qualitative methods enable a realistic portrayal of the rich diversity of human experiences, which are difficult to capture in purely quantitative studies.

Although several phenomenological studies are conducted using quantitative methods, many authors agree that the epistemological stance in this type of research warrants careful attention [46,95,96]. For this reason, Dibsdall et al. [46] argue that qualitative research provides more meaning and understanding than statistical analysis when investigating consumer behaviors. Therefore, it is important to recognize that the epistemological question remains open for this type of research.

In addition to the research directions listed above, an analysis of different studies has allowed the chronological summarization of other potential research avenues in the field of artificial meat consumption (see Table 3).

**Table 3 foods-15-00034-t003:** Chronological summary of future research identified in the articles.

In 2015
Hocquette et al. [97] suggested that a potential research direction would be to better understand the long-term diversity of consumer opinions regarding the acceptability of artificial meat. Verbeke et al. [19], for their part, recommended continuing studies on personal and environmental determinants, particularly personal motivations and the effects of information that may shape perceptions, expectations, and the likelihood of consumer acceptance or rejection of cultured meat. These authors also advocate examining the influence of beliefs, ethnicity, acculturation, or specific perceptions on the acceptance of artificial meat. Furthermore, Laestadius [72] proposed new comparative research on the acceptability of cultured meat relative to other meat alternatives, as well as additional studies on consumer perceptions of cultured meat, focusing on the establishment of a public dialogue regarding cultured meat.
In 2019
The work of Bryant et al. [31] suggested that each country should explore consumer acceptance of plant-based and cultured meat, given that most surveys differ in question wording, response options, and terminology. They recommend conducting studies on the role of regulations in individuals’ judgments regarding food safety. For their part, Mancini and Antonioli [68] suggest the need to better understand whether a positive perception predicts a willingness to try or purchase cultured meat. They also recommend analyzing consumer perceptions and expectations, particularly in countries where culture, tradition, and/or religion might hinder the acceptance of cultured meat.
In 2020
According to Mancini and Antonioli [69], researchers should conduct studies on the impact of information on consumer acceptance of cultured meat.
In 2023
For Kouarfaté and Durif [2], it would be interesting to explore the influence of stakeholders on the acceptance of cultured meat and for Cornelissen and Piqueras-Fiszman [26], future studies should focus on the segmentation factors of these consumers.

## 7. Theoretical and Managerial Contributions

This article has the merit of proposing a register for the different axiological values of cultured meat. Moreover, it has also identified and provided a list of amplifiers of these values, highlighting their role. This research has further demonstrated the impact of the axiological perspective of artificial meat on consumer perception and on the acceptance of cultured meat.

By exploring the notion of axiological value in the analysis of consumption behavior, this study enables the adaptation of communication (advertising messages) to emphasize the benefits of cultured meat. Indeed, the register of axiological values identified in this work allows agri-food companies to deploy specific advertising messages capable of enhancing consumer perception based on each of these values.

The research also revealed that co-production particularly “dogmatic co-production” like all other amplifiers, is a factor that enhances the value of cultured meat and its perception in consumers’ minds. Such co-production gives consumers a sense of control over the production of this meat, thereby improving their satisfaction while giving them a more significant role in decision-making. Consequently, participation in production can strengthen their relationship with cultured meat by creating a sense of proximity and reducing feelings of exploitation [98].

Finally, by giving consumers the opportunity to contribute to the production of cultured meat (through their opinions and suggestions), the industry can better align the final product with consumer needs, thereby improving satisfaction, reinforcing engagement, and creating value. This, in turn, further enhances the axiological value of cultured meat. Nevertheless, companies should ensure that collaboration between consumers and cultured meat producers is appropriately managed, in accordance with the laws of each country.

## 8. Limitations of the Research

The main limitation of this study is certainly related to the absence of a phenomenological analysis in the field to take into account consumer opinions. This lack of analysis highlights the subjectivity of the researchers, which is associated with the qualitative research paradigm. However, since this is a systematic literature review, the studies used in this work had already been researched with consumers, and therefore, the relevance of this study cannot be called into question. The grouping of the axiological values of cultured meat was carried out based on 54 articles. This grouping cannot therefore be complete and exhaustive. Indeed, other authors could conduct further studies either to confirm these axiological groupings or to improve them. In this article, specific tools for assessing the quality of the articles were not used. However, the article quality assessment tools integrated into the ‘Sofia Library’ served as instruments for evaluating this quality. Thus, the Sofia Library provides the possibility of automatically selecting peer-reviewed articles. Furthermore, almost all of the selected articles are published in journals included in the ranking of scientific journals and countries, which attests to the quality of these articles. Finally, these quality criteria have already been used by other researchers, including Kouarfaté and Durif [2], Abd Hanan et al. [24], and Ji and Lee [54]. Another limitation of this study concerns the potential bias related to the exhaustiveness of the selection of studies based on the requirement for exact keyword matches in article titles at the time of selection. However, this requirement helps improve the precision and quality of the articles retained with regard to the topic addressed in the research. Moreover, several authors have also used this requirement of exact keyword matching in titles during their selection process [2,21,24,54]. In addition, despite this requirement, several articles were identified and selected (54 articles, which is a more than sufficient number), since some studies included fewer articles in their analysis for their systematic literature review (see Bryant and Barnett, [21]; Abd Hanan et al., [24]; and Ji and Lee, [54]).

## 9. Conclusions

The axiological values of cultured meat, analyzed from the perspective of phenomenological research methods on perception and consumption, made it possible to establish the register of axiological values as well as their amplifiers. By applying a systematic methodology, this analysis of the articles was conducted using NVIVO12 software, which revealed that the axiological values of cultured meat each influence both consumer perception and acceptance of this meat. Furthermore, by identifying the amplifiers of these different axiological values, the study showed that “dogmatic” co-production (religious or cultural co-production) could significantly improve consumer perceptions and acceptance of cultured meat. Finally, although this study has limitations, it has the merit of summarizing various research avenues and providing original value and contribution, notably by improving communication strategies surrounding cultured meat based on axiological values and amplifiers.

## Figures and Tables

**Figure 2 foods-15-00034-f002:**
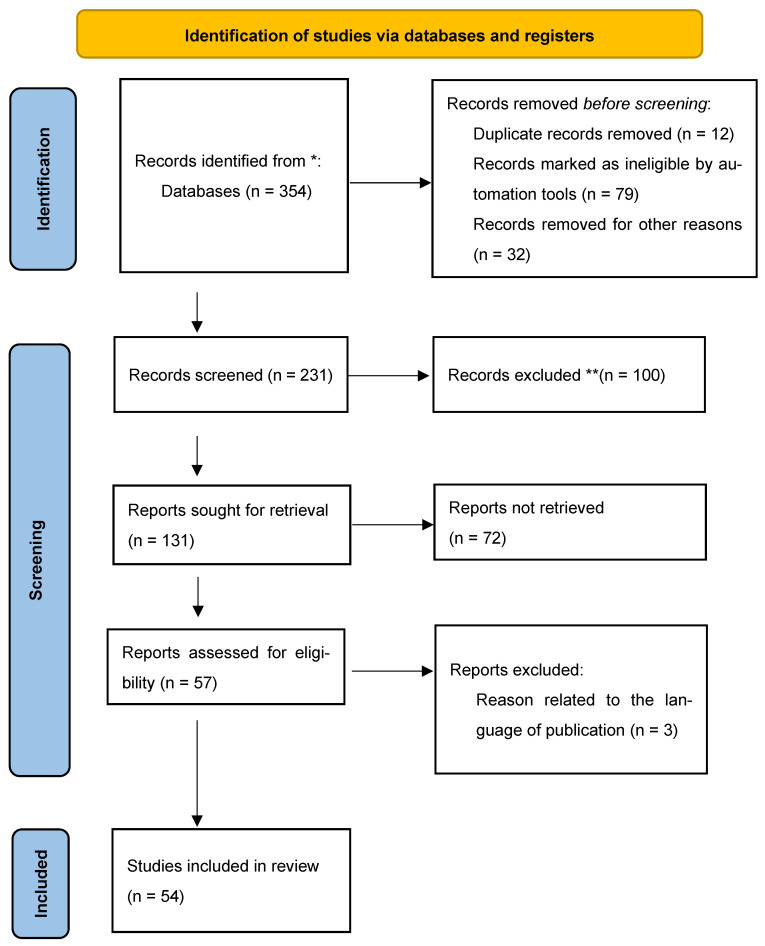
Research protocol based on PRISMA 2020 flow diagram for new systematic reviews which included searches of databases and registers only [55]. * Consider, if feasible to do so, reporting the number of records identified from each database or register searched (rather than the total number across all databases/registers). ** If automation tools were used, indicate how many records were excluded by a human and how many were excluded by automation tools [55]. This work (diagram) is licensed under CC BY 4.0. To view a copy of this license, visit https://creativecommons.org/licenses/by/4.0/ (accessed on 30 August 2025).

**Figure 3 foods-15-00034-f003:**
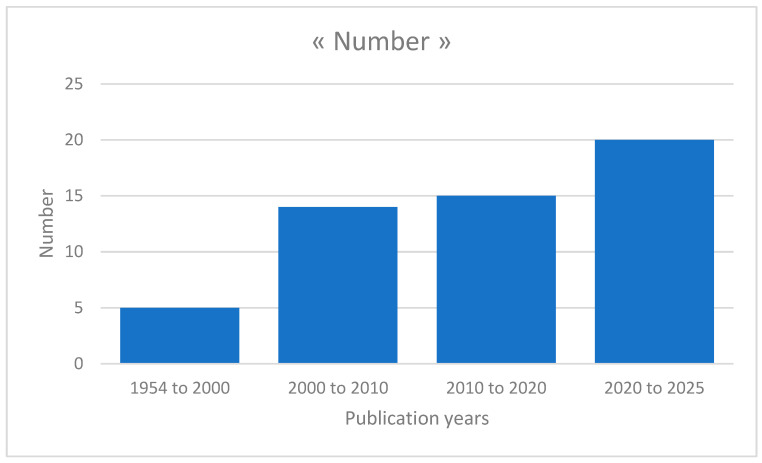
The number of articles per year of publication.

**Figure 4 foods-15-00034-f004:**
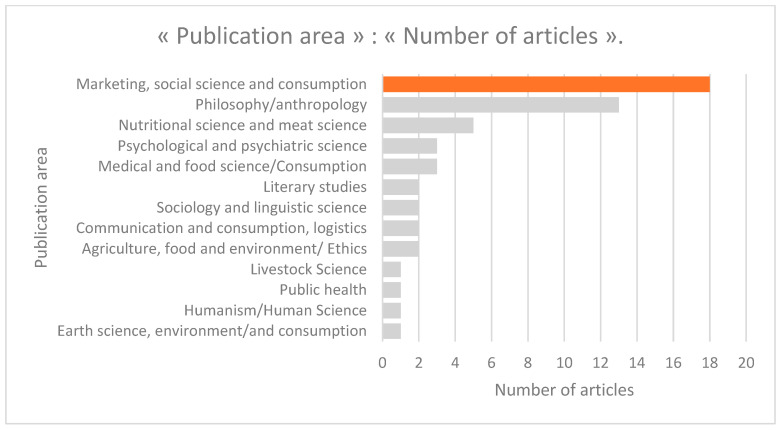
Article publication area.

**Table 1 foods-15-00034-t001:** Inclusion and exclusion criteria used for article selection.

Integration or Inclusion Criterion	Exclusion Criterion
Full-text article published in a peer-reviewed journal Articles that accurately include the keywords in their title Across all studies (quantitative and/or qualitative) Downloadable articles Language: English, French	Sources not peer-reviewed Articles whose titles do not contain keywords. Incomplete articles and elimination of duplicates Studies published in languages other than French and English.

**Table 2 foods-15-00034-t002:** Article selection process.

Keywords	Phenomenology of Perception	Axiology of Consumption	Axiology	Cultured Meat and Perception	Artificial Meat and Perception	In Vitro Meat Perception	Phenomenology and Food	Axiology and Communication
Exact expression in the title of the articles	126	35	67	59	24	9	22	12
Peer-reviewed papers	89	12	32	46	20	7	17	8
Downloadable articles	6	12	32	42	14	7	12	6
Incomplete articles and elimination of duplicates	3	9	8	38	12	7	9	5
Full text (Articles selected for our analysis)	3	5	6	19	5	4	7	5

## Data Availability

No new data were created or analyzed in this study.

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
