# Peer review of "Axiology of Cultured Meat and Consumer Perception: An Analysis Based on the Phenomenology of Perception"

_foods, 2025, doi:10.3390/foods15010034_

Round 1

Reviewer 1 Report

Comments and Suggestions for Authors

This article addresses an important and innovative topic: combining axiology, consumer perception, and phenomenology in the analysis of in vitro-grown meat. This is undoubtedly an interesting theoretical contribution. However, the article requires several refinements.

  1. Please ensure that keywords do not repeat the article title.
  2. Add a full search strategy (dates, queries, filters).
  3. Use a quality assessment tool/
  4. Add a table with the rating of each article.
  5. Please clearly explain what elements of phenomenology will be used and why this is appropriate in the literature review.
  6. The theoretical construction of "axiological values" is imprecise. Axiological categories were arbitrarily created based on very general literature (primarily philosophical) and then assigned to meat research. There is no description of the procedure by which the authors identified the seven value categories. There are no operational definitions of individual values.
  7. The results are descriptive and replicate content from the literature, but do not synthesize the data. There are no coding statistics (e.g., the number of articles assigned to a given value). Sample quotes or text excerpts, which form the basis for interpretation, are not provided.
  8. The authors claim that there are no axiological reviews on cultured meat, but they do not show which areas the gap concerns.
  9. The current restrictions are very short and do not cover key risks .

Author Response

For research article: Axiology of Cultured Meat and Consumer Perception: An Analysis Based on the Phenomenology of Perception

Response to Reviewer 1 Comments

1. Summary

Comments and Suggestions for the Authors
This article addresses an important and innovative topic: the combination of axiology, consumer perception, and phenomenology in the analysis of in-vitro cultivated meat. It is undeniably an interesting theoretical contribution. However, the article requires several improvements.

  1. Please make sure that the keywords do not repeat the title of the article.
    Response: According to the recommendation of the journal Foods, the title of the article should contain 2 to 3 keywords used in the manuscript. However, if agreeable to you, we can modify the title as follows: “Improving consumer perceptions through the axiological values of cultured meat.” We are also open to your suggestions for a new title.
  2. Add a complete search strategy (dates, queries, filters).
    Response: We have added a complete search strategy, including dates, queries, and filters, in the section “3.3. Evaluation of the Quality of the Selected Studies.” (See the paragraph highlighted in green.)
  3. Use a quality assessment tool.
    Response: The quality assessment tool used in this paper is the one integrated in the “Sofia Library,” which is the standardized library system for all Canadian universities. The Sofia Library automatically allows the selection of peer-reviewed articles.
    However, to fully address your request, we have added an additional paragraph in the methodology section providing more details on the quality criteria of the articles included in this paper. Furthermore, as evidence of article quality, we added a short paragraph highlighted in green in section “4.1. Results of the Descriptive Analysis (Profile of the Analyzed Articles).” Here is the paragraph:
    “Almost all selected articles (51/54) are published in journals listed in the ranking of scientific journals and countries, including 32 in the first quartile (Q1), 5 in the second quartile (Q2), 7 in the third quartile (Q3), and 7 in the fourth quartile (Q4), which attests to their quality.”
    Moreover, this article follows the methodology recommended by Tranfield et al. (2003), which does not require specific quality-assessment tools. However, the quality of the selected documents is based on the fact that they are peer-reviewed. We have mentioned this in the methodology. The same methodology has been used by several researchers in systematic literature reviews (e.g., Kouarfaté and Durif, 2023a; Abd Hanan et al., 2025; Ji and Lee, 2026).
  4. Add a table with the evaluation of each article.
    Response: We have added a table in the appendix titled “Appendix 2 Table 5: Evaluation of Scientific Articles.” However, this evaluation is based on the fact that all included articles were peer-reviewed and published in journals listed in scientific journal rankings.
  5. Clearly explain which elements of phenomenology will be used and why this is appropriate for the literature review.
    Response: This has been addressed. We clearly explained the elements of phenomenology that will be used and why they are appropriate for the literature review by adding a paragraph (see the section highlighted in green in “2. Literature Review”).
  6. The theoretical construction of “axiological values” is imprecise. The axiological categories were created arbitrarily from a very general (mainly philosophical) literature and then applied to meat research. There is no description of the procedure used to identify the seven value categories, and no operational definitions of the individual values.
    Response: This has been addressed. We added a section in the methodology explaining the coding of themes corresponding to axiological values. Additionally, we included several paragraphs in section “3.5. Data Analysis and Interpretation” (see the green-highlighted parts).
  7. The results are descriptive and replicate the content of the literature without synthesizing the data. There are no coding statistics (e.g., number of articles assigned to a given value). No quotations or text excerpts supporting the interpretations are provided.
    Response: This has been addressed. We added quotations and excerpts serving as the basis for interpretation, as requested (see the green-highlighted sections in the subsections of section 4.2).
    For coding statistics, they can be found in section “4.1. Results of the Descriptive Analysis (Profile of the Analyzed Articles)” as well as in Figures 1 and 2.
  8. The authors claim that there are no axiological reviews on cultured meat, but they do not indicate which fields are concerned by this gap.
    Response: The field concerned by this gap is mentioned in the sentence highlighted in green in the literature review (see paragraph 3 of section “2. Literature Review”).
  9. The current limitations are very brief and do not cover key risks.
    Response: This has been addressed. We added additional limitations (see the green-highlighted parts in section “8. Limitations of the Research”).

Reviewer 2 Report

Comments and Suggestions for Authors

The manuscript tackles a highly relevant topic related to consumer acceptance of cultured meat through an innovative conceptual approach combining axiology and phenomenology. While the topic and intentions are valuable, the manuscript requires substantial improvements in methodological transparency, conceptual refinement, and writing quality before it can be considered for publication.

Major Comments

- Requiring exact keyword matches in article titles (Table 2) considerably restricts the number of eligible studies and introduces selection bias. Broader search fields (e.g., title/abstract/keywords) and fully documented search syntax are needed to ensure reproducibility and robustness. How will the authors address potential bias and improve the comprehensiveness of study selection?

- Although NVivo-based coding is mentioned, no details are provided on the development of the coding framework, coder involvement, reliability procedures, or sample coded extracts. Greater methodological transparency is necessary to strengthen the credibility of the thematic analysis.

- Several axiological categories appear conceptually overlapping, and the distinctions between hermeneutic and reputational values are not well grounded in established frameworks within food perception and innovation adoption research. A clearer theoretical justification is required.

- The concept of “dogmatic co-production” is insufficiently evidenced. While the idea is theoretically interesting, the current justification is largely inferential rather than clearly grounded in empirical findings from the included studies. On what specific evidence does this concept rely within the reviewed literature?

- The Discussion section adopts a predominantly confirmatory perspective, with limited engagement in contradictions, uncertainties, or boundaries of the evidence. A more critical and analytical approach is needed to enhance the scholarly contribution.

Minor Comments

- Clarify hypotheses derived from conceptual model (Figure 1).

- Improve consistency of terminology regarding types of meat alternatives.

- Review English phrasing to reduce redundancy and enhance clarity.

- Improve alignment between tables/figures and text.

Issues with Manuscript Formatting and Presentation

- Several images, charts, and tables appear with low resolution, which can hinder interpretation of details (e.g., PRISMA diagram and publication distribution graphs).

- Several images appear to be screenshots taken directly from a word processor with active spell-check, resulting in red underline marks visible under words in the figures.

Author Response

For research article: Axiology of Cultured Meat and Consumer Perception: An Analysis Based on the Phenomenology of Perception

Response to Reviewer 2 Comments

1. Summary

Comments and Suggestions for the Authors

The manuscript tackles a highly relevant topic related to consumer acceptance of cultured meat through an innovative conceptual approach combining axiology and phenomenology. While the topic and intentions are valuable, the manuscript requires substantial improvements in methodological transparency, conceptual refinement, and writing quality before it can be considered for publication.

Response: All corrections have been made.

Major Comments

  • Requiring exact keyword matches in article titles (Table 2) considerably limits the number of eligible studies and introduces selection bias. Broader search fields (e.g., title/abstract/keywords) and fully documented search syntax are needed to ensure reproducibility and robustness. How do the authors intend to address potential bias and improve the comprehensiveness of the study selection?
    Response: Corrections have been made.

Regarding reproducibility and robustness, we have corrected and documented the search syntax (see the sections highlighted in green in 3.3 and 3.5).
Concerning the requirement for exact keyword matches in article titles, this helps improve the precision and relevance of the selected studies with respect to the topic addressed. Several authors have used this requirement for exact keyword matching in titles (Bryant & Barnett, 2018; Kouarfaté & Durif, 2023; Abd Hanan et al., 2025; Ji & Lee, 2025). Moreover, despite this requirement, we were able to include 54 articles, which is more than sufficient for a systematic literature review, especially given that some studies retained far fewer articles (see Bryant & Barnett, 2018; Abd Hanan et al., 2025; Ji & Lee, 2025).
However, to address your request, we have added this point as a limitation (see the section “8. Limitations of the Research”).

  • Although NVivo coding is mentioned, no details are provided regarding the development of the coding framework, coder involvement, reliability procedures, or illustrative coded excerpts. Greater methodological transparency is needed to strengthen the credibility of the thematic analysis.
    Response: All corrections have been made. We added details regarding the development of the coding framework, coder involvement, and reliability procedures (see the green-highlighted sections in 3.3 and 3.5).
    For illustrative coded excerpts, we added them in each subsection of “4.2. Register of Axiological Values of Cultured Meat” (see the green-highlighted portions throughout section 4.2). These corrections improve methodological transparency and strengthen the credibility of the thematic analysis.
  • Several axiological categories appear conceptually overlapping, and the distinctions between hermeneutic and reputational values are not solidly grounded in established frameworks on food perception and innovation adoption. A clearer theoretical justification is required.
    Response: Clearer justifications regarding the distinction between hermeneutic and reputational values are highlighted in yellow within each corresponding section. The distinction is based on the fact that “reputation stems from stakeholders through aspects such as governance, social responsibility, and excellence, which reinforce trust and collective esteem,” whereas hermeneutic values arise from consumers’ perceptions based on the initial information they receive about cultured meat (e.g., curiosity or food neophobia).
  • The concept of “dogmatic co-production” is not sufficiently substantiated. Although the idea is theoretically interesting, the current justification relies largely on inference rather than on clear empirical evidence from the reviewed studies. On what specific evidence does this concept rely within the analyzed literature?
    Response: In accordance with your request, we added several empirical excerpts from the examined studies to support the concept of “dogmatic co-production” (see the green-highlighted excerpts in the subsection “The co-production of value” in section 4.2.8).
  • The Discussion section adopts a predominantly confirmatory perspective, with limited engagement with contradictions, uncertainties, or data limitations. A more critical and analytical approach is necessary to strengthen the scientific contribution.
    Response: The confirmatory nature of the discussion reflects the fact that the findings across the reviewed studies are consistent with our own. However, following your request, we added paragraphs in the Discussion section to adopt a more critical and analytical approach, thereby strengthening the scientific contribution (see the green-highlighted parts in section “5. Discussion”).

Minor Comments

  • Clarify the hypotheses derived from the conceptual model (Figure 1).
    Response: This has been done. hypotheses derived from the conceptual model have been clarified (see the green-highlighted section under “Conceptual Framework”).
  • Improve the consistency of terminology relating to types of meat alternatives.
    Response: This has been addressed. Additional precision has been added (see the yellow-highlighted part in section “2. Literature Review”).
  • Review the English phrasing to reduce redundancies and improve clarity.
    Response: We will pay for professional English editing services from MDPI Editing.
  • Better align the tables/figures with the text.
    Response: Yes, after acceptance, we will use MDPI Editing services to ensure proper formatting of tables and figures according to Foods journal guidelines.

Formatting and Presentation Issues

  • Several images, charts, and tables appear in low resolution, which may hinder interpretation of details (e.g., PRISMA diagram and publication distribution graphs).
    Response: The PRISMA diagram is the recommended version we used. However, all figures and tables will be adjusted and corrected to meet Foods journal formatting guidelines through MDPI Editing services.
  • Several images appear to be screenshots from a word-processing tool with an active spellchecker, resulting in visible red underlines beneath some words.
    Response: All such issues will be corrected using MDPI Editing services. After acceptance of the paper, we will submit the document to MDPI Editing.

Reviewer 3 Report

Comments and Suggestions for Authors

‘Fifty-four peer-reviewed qualitative and quantitative studies, identified through the Libraries Worldwide database’ – this is not a proper database in terms of content filtering, classification, and ranking. I suggest you use the Web of Science and Scopus instead. What quality tools did you use in article selection? E.g., evidence synthesis screening software and reference and review management tools that can be harnessed include Abstrackr (for machine-learning-based abstract organization), AXIS (for cross-sectional study quality assessment), CADIMA (for evidence-based decision making support), CASP (for thematic synthesized evidence quality appraisal), Catchii (for article relevance assessment), R package and Shiny app citationchaser (for backward and forward citation tracking), Eppi-Reviewer (for machine learning-based data clustering), JBI SUMARI (for published content trustworthiness evaluation), METAGEAR package for R (for text mining and critical analysis), MMAT (for design type determination), PICO Portal (for collaborative knowledge quality assessment), SluRp (for data clustering and categorization), and SWIFT-Active Screener (for document classification and prioritization), and Systematic Review Accelerator (for search string duplicate exclusion). ‘However, several previous studies’ – why ‘however’? This paragraph does not contradict the previous one. "phenomenological inquiry should be considered a vehicle and method for accessing the food experience", "to examine what is good or worthy of esteem, exploring the nature, origin, and different types of values (moral, aesthetic, epistemic, etc.), as well as how to measure and compare them" – include pages for the quotes, but anyway such ideas can be easily paraphrased. There are several such instances. ‘several studies have shown that cultured meat could meet humanity's growing needs [3] while reducing greenhouse gas emissions associated with conventional meat production’ – ‘studies’, but you mention only one source. Also, the claim after [3] belongs to you or to the cited authors? ‘from an animal [3,19]’ – avoid citing the same source twice in a row. There are several such instances. ‘faces significant obstacles [20]’ – what are those obstacles? Be always concrete. ‘a survey conducted in Canada in 2020 revealed that only 22% of the population said they were willing to try it [20]’ – quite old data. Maybe things have changed. ‘For example, [4,18,23,24] identified the determinants that influence consumer attitudes toward this meat’ – what are those determinants? ‘For example, Golev [44] studied the cognitive and communicative axiology of modern virtual educational communication’ – and what were the results? ‘This is why several authors such as [11], [49], [49], [47] and [48] have used this method in their various research on consumer consumption and perceptions’ – you are supposed to clarify the positions for all of them, not only for some. Figure 1 is unclear. ‘The analysis of the work of several authors such as [2,14,21,31,46,59,62,72–74]’ – develop and clarify the specific contribution of each cited source. ‘According to [2,56], the values of cultured meat are associated with environmental and public health motivations, unnaturalness, regulatory considerations, and neophobia, which influence consumer perceptions of this type of meat [56]’ – you first support this claim by 2 sources, and then only by one. When does the contribution of both end? The same here: ‘Additionally, according to Teklebrhan [83], religious and sociocultural taboos are key variables affecting meat preferences and consumption within a given population [83,84]’. Graph 1 is in French. ‘his analysis of the articles was conducted using NVIVO12 software’ – it is unclear how this software was used. As the topic is very hot, with continuous technological development integration, I would have expected that most cited sources be from the past 2 years, and mainly from WoS SSCI/SCIE journals.

Author Response

For research article: Axiology of Cultured Meat and Consumer Perception: An Analysis Based on the Phenomenology of Perception

Response to Reviewer 3 Comments

1. Summary

Comments and suggestions for the authors

“Fifty-four peer-reviewed qualitative and quantitative studies, identified through the Libraries Worldwide database” — this is not an adequate database in terms of content filtering, classification, and ranking. I suggest using Web of Science and Scopus instead.
Response: In this study, all these databases were taken into account (including Web of Science and Scopus) in order to include scientific studies from all fields and disciplines. Indeed, the literature search was conducted via the Sofia library. It is the “unified” library of all universities across all Canadian provinces. This library contains all databases, including Web of Science and Scopus. Likewise, articles were automatically selected using the “advanced search” function of the Sofia library by choosing the “Libraries Worldwide” option. Regarding query settings and article filtering, automated filtering tools were used within Sofia. For instance, for databases, we selected the “all databases” option; for article quality, the “peer-reviewed articles” option; for duplicates, the “remove duplicates” option. For the “content type”, the “Full Text” and “Scholarly Publications” options were selected. For publication year, the “all years” option was selected up to August 30, 2025.
The retrieved articles were jointly evaluated by the entire team, which included two specialists in qualitative data analysis and the lead author of this article. Disagreements were resolved by consensus, for example by jointly evaluating the methodology, study design, or results. The decision to include articles from all disciplines reflects the multidisciplinary nature of this work.

Which quality assessment tools did you use to select the articles? For example, screening tools for evidence synthesis and reference-management software include: Abstrackr, AXIS, CADIMA, CASP, Catchii, R packages and Shiny citationchaser, Eppi-Reviewer, JBI SUMARI, METAGEAR for R, MMAT, PICO Portal, SluRp, SWIFT-Active Screener, and the Systematic Review Accelerator.
Response: The quality-assessment tool used in this paper is the one integrated into the Sofia Library, as it allows automatic filtering of peer-reviewed articles published in journals that appear in recognized journal and country rankings. Thus, to ensure sufficient quality, only peer-reviewed articles were retained. Such articles undergo rigorous academic review processes ensuring methodological soundness, follow standardized methodologies facilitating comparison and synthesis, and are generally less exposed to commercial or political biases than other types of literature such as industry reports or policy documents.
These quality criteria have already been used by other researchers, notably Bryant and Barnett (2018), Kouarfaté and Durif (2023), Abd Hanan et al. (2025), and Ji and Lee (2025).
Furthermore, to ensure accessibility, quality, and relevance, only publications written in English or French were included.

However, to fully address your request, we added an additional paragraph in the methodology section explaining the quality criteria applied. Moreover, to demonstrate article quality, a short paragraph highlighted in green was inserted in Section “4.1. Results of the Descriptive Analysis”:

“Almost all of the selected articles (51/54) are published in journals listed in the scientific and country ranking systems: 32 in the first quartile (Q1), 5 in the second quartile (Q2), 7 in the third quartile (Q3), and 7 in the fourth quartile (Q4), which attests to the quality of these articles.”

In addition, for this paper we used the methodology recommended by Tranfield et al. (2003). This methodology does not require the use of specific quality-assessment tools. Nevertheless, the quality of the selected documents is supported by the fact that they are peer-reviewed. We have indicated this in the methodology section. This same methodology has been used by many scholars in systematic literature reviews (e.g., Kouarfaté & Durif, 2023a; Abd Hanan et al., 2025; Ji & Lee, 2026).
Furthermore, to address your concern, we added this point as a study limitation (highlighted in green in Section 8. Limitations of the Research).

“However, several previous studies” — why “however”? This paragraph does not contradict the previous one.
Response: We corrected the sentence (see the highlighted green text in the second paragraph of Section 2. Literature Review).

“phenomenological inquiry should be considered a vehicle and method for accessing the food experience”
Response: Done. We added the page number for this citation (highlighted in yellow in Section 2. Literature Review).

“to examine what is good or worthy of esteem…” — please indicate the page numbers for these citations; however, these ideas can easily be reformulated. Several similar cases appear in the text.
Response: Done. We reformulated the sentence accordingly (highlighted in green in Section 1. Introduction).

“several studies have shown that cultured meat could meet humanity's growing needs [3]…” — you mention “several studies”, but only one source is cited. Also, does the statement after [3] belong to you or to the cited authors?
“from an animal [3,19]” — avoid citing the same source twice in a row.
Response: Done. We added a second reference and corrected the sentence (highlighted in green in the first paragraph of Section 2. Literature Review).

“faces significant obstacles [20]” — what obstacles? Always be concrete.
Response: Done. We clarified the obstacles (highlighted in green in the second sentence of the first paragraph).

“a survey conducted in Canada in 2020… only 22%… willing to try it [20]” — rather old data; the situation may have changed.
Response: You are absolutely right that the situation has likely evolved. However, no new studies have been published on Canadians’ willingness to try cultured meat.

“For example, [4,18,23,24] identified the determinants…” — what are these determinants?
Response: We provided further details about the determinants (highlighted in red in the second paragraph of Section 2. Literature Review).

“For example, Golev [44] studied the cognitive and communicative axiology…” — what were the results?
Response: Done. We added the study results (highlighted in green in the fourth paragraph of Section 2. Literature Review).

“This is why several authors such as [11], [49], [49], [47], [48]…” — you must clarify the contribution of each author, not only some of them.
Response: Done. We clarified each author's position by adding the missing details (highlighted in red in the fifth paragraph of Section 2. Literature Review).

Figure 1 is unclear.
Response: Figure 1 can be enlarged. We also requested MDPI editing services to improve tables and figures.

“The analysis of the work of several authors such as [2,14,21,31,46,59,62,72–74]” — develop and clarify each source’s specific contribution.
Response: We developed and clarified each source’s contribution (highlighted in yellow in Section 4.2.8. The Axiological Value Amplifiers of Cultured Meat).

“According to [2,56], the values of cultured meat are associated with…” — you first support the statement with two sources, then only one. Clarify when each contribution begins and ends.
Response: Done. We added the second source (highlighted in yellow in Section 5. Discussion).

Same issue here:
“Additionally, according to Teklebrhan [83], religious and sociocultural taboos…”
Response: Done. We added the second source (highlighted in yellow in Section 5. Discussion).

Graph 1 is in French.
Response: Done. The graph is now in English. MDPI will ensure final language corrections before publication.

“this analysis of the articles was conducted using NVIVO12 software” — it is not clear how the software was used.
Response: We added a full paragraph explaining how NVivo 12 was used (highlighted in green in Section 3.5).

Given that the topic is very current… one would expect that most cited sources would date from the last two years and come from WoS SSCI/SCIE journals.
Response: We cited many recent articles in this paper, including some from 2025. However, to properly cover the literature, certain “classic” articles were also included.

Round 2

Reviewer 1 Report

Comments and Suggestions for Authors

The authors revised the article according to the reviewer's suggestion.

Reviewer 2 Report

Comments and Suggestions for Authors

The authors have adequately addressed the reviewers’ comments and introduced meaningful improvements to the manuscript, particularly by strengthening methodological transparency, clarifying the axiological framework, and better grounding the concept of co-production within the existing literature. While some minor limitations remain, they are appropriately acknowledged and do not detract from the overall quality or contribution of the study. Overall, the manuscript is now suitable for publication and may be accepted, subject only to minor editorial adjustments.